# Hybrid Dihydropyrimidinones Targeting AKT Signaling: Antitumor Activity in Hormone-Dependent 2D and 3D Cancer Models

**DOI:** 10.3390/pharmaceutics17111470

**Published:** 2025-11-14

**Authors:** Amanda Helena Tejada, Samuel José Santos, Gabriel Tofolli Lobo, Abu-Bakr Adetayo Ariwoola, Aryel José Alves Bezerra, Giulia Rodrigues Stringhetta, Izabela Natalia Faria Gomes, Luciane Sussuchi da Silva, Rui Manuel V. Reis, Daniel D’Almeida Preto, Dennis Russowsky, Renato José Silva-Oliveira

**Affiliations:** 1Molecular Oncology Research Center, Barretos Cancer Hospital, Antenor Duarte Villela, 1331, Barretos 14784-400, SP, Brazil; amandahtejada@gmail.com (A.H.T.); gabrieltoffolilobo@icloud.com (G.T.L.); tayoariwoola@gmail.com (A.-B.A.A.); aryelbezerra@gmail.com (A.J.A.B.); giuliarodriguesstringhetta@gmail.com (G.R.S.); izabela.faria.tk@hotmail.com (I.N.F.G.); lsussuchi@gmail.com (L.S.d.S.); ruireis.hcb@gmail.com (R.M.V.R.); 2Laboratório de Sínteses Orgânicas, Instituto de Química, Universidade Federal do Rio Grande do Sul, Porto Alegre 90010-150, RS, Brazil; 3Life and Health Sciences Research Institute (ICVS), School of Medicine, University of Minho, 4704-553 Braga, Portugal; 4ICVS/3B’s-PT Government Associate Laboratory, 4710-057 Braga/4806-909 Guimarães, Portugal; 5Clinical Oncology Department, Barretos Cancer Hospital, Antenor Duarte Villela, 1331, Barretos 14784-400, SP, Brazil; ddpreto@gmail.com; 6Faculty of Health Sciences of Barretos Dr. Paulo Prata–FACISB, Barretos 14785-002, SP, Brazil

**Keywords:** hybrid compounds, AKT, apoptosis, hormone therapy, resistance, breast cancer, prostate cancer, ovarian cancer

## Abstract

**Background/Objectives:** The development of effective oncologic therapies with fewer adverse effects is often limited by the intrinsic and acquired resistance of tumor cells. Hybrid molecules, rationally designed to combine different pharmacophores, represent a promising strategy by providing synergistic effects, dose reduction, and a lower risk of resistance. In this study, the antitumor potential and mechanisms of action of 22 novel hybrid compounds derived from xanthene and pyran scaffolds (SJ022–SJ103) were investigated. The hybrids were initially evaluated through in vitro screening in four breast, three ovarian, and two prostate cancer cell lines, followed by the selection of T-47D, OVCAR-3, and LNCaP cells for detailed assays assessing cytotoxicity, apoptosis, cell cycle distribution, DNA damage, caspase-3/7 activity, morphology, and PI3K/AKT/mTOR pathway modulation. **Methods:** Cytotoxicity assays were performed in the selected cell lines, while mechanistic studies included apoptosis and cell cycle analysis by flow cytometry, γH2AX detection, Western blotting for PI3K/AKT/mTOR pathway proteins, and 3D spheroid assays. Combinatorial effects with hormone therapies (tamoxifen, fulvestrant, and letrozole) and the AKT inhibitor MK2206 were evaluated. AKT silencing by esiRNA and molecular docking was performed to confirm target engagement. **Results:** SJ028 demonstrated broad activity across all tested cell lines, whereas SJ064 and SJ078 exhibited higher selectivity. Treatments induced apoptosis, S/G2-M arrest, and DNA damage, accompanied by decreased phospho-AKT levels and stable PI3K and mTOR expression. In 3D models, the hybrids increased caspase-3/7 activity and necrotic core expansion. Co-administration with hormone therapies resulted in synergistic effects in breast and ovarian cancer cells, reducing IC_50_ values by more than 50% in both parental and resistant models, while combinations with MK2206 were antagonistic across all tumor subtypes. AKT silencing abrogated cytotoxicity, and docking confirmed SJ028 binding to AKT. **Conclusions:** Xanthene- and pyran-based hybrids—particularly SJ028, SJ064, and SJ078—showed strong antitumor activity through apoptosis induction, cell cycle arrest, and PI3K/AKT pathway modulation. Their preserved efficacy in resistant models and synergistic interactions with hormone therapies contrasted with the antagonism observed with AKT inhibition, highlighting their potential as promising candidates for the treatment of hormone-responsive and -resistant cancers.

## 1. Introduction

Hormone-dependent tumors, including breast, prostate, and ovarian cancers, are particularly influenced by sex hormones [1]. In breast cancer, estrogen (ER), progesterone (PR), and HER2 receptors define molecular subtypes and guide therapy [2,3]. Approximately 75% of breast cancers are hormone receptor positive. However, both intrinsic and acquired resistance to endocrine therapy pose major clinical challenges, contributing significantly to recurrence after initial treatment and to disease progression in the metastatic stages [4]. In the early stages, most prostate cancers are androgen-dependent, and androgen deprivation therapy (ADT) is the standard of care. Nonetheless, over time, many patients develop castration-resistant prostate cancer, marked by continued tumor growth despite castration-level androgen concentrations [5,6]. Ovarian cancer ranks as the fifth leading cause of cancer-related death among women. Gonadotropin hormone-releasing hormone (GnRH) promotes pituitary release of luteinizing hormone (LH) and follicle-stimulating hormone (FSH), which can in turn drive ovarian cancer cell proliferation. Suppression of gonadotropin secretion by superactive GnRH analogs may therefore offer therapeutic benefit [7]. While estrogen signaling contributes to ovarian tumorigenesis and therapeutic response [1]. Endocrine therapies, such as selective estrogen receptor modulators (SERMs), aromatase inhibitors, and selective estrogen receptor degraders (SERDs), are central to managing hormone-dependent cancers [2]. However, resistance remains a major clinical challenge, mediated by mutations in hormone receptors [8] dysregulation of co-regulators, and activation of alternative proliferative pathways, including PI3K/AKT/mTOR and MAPK [6]. Similar mechanisms underlie resistance in castration-resistant prostate cancer [3,6,9] and in subsets of ovarian tumors [10], limiting the efficacy of hormonal interventions.

Natural and synthetic heterocyclic compounds, particularly xanthene and pyran derivatives, have demonstrated cytotoxic, antiproliferative, and pro-apoptotic effects across breast, prostate, and ovarian cancer models [11]. Hybrid molecules combining xanthene and pyran scaffolds, as 5,6-dimethylxanthone-4-acetic acid (D) and pyranoxanthone (P) hybrids, have the potential to target multiple signaling pathways simultaneously, overcoming resistance mechanisms and enhancing therapeutic efficacy [12]. Therefore, the development and evaluation of xanthene-pyran hybrids represent a rational and innovative strategy to advance multitarget therapies for hormone-dependent tumors, addressing critical challenges in overcoming endocrine resistance and improving clinical outcomes [10,13].

The present study aimed to systematically evaluate a library of 22 xanthene–dihydropyrimidinone hybrid compounds for their antitumor potential, elucidate their putative cellular targets, and characterize their mechanisms of action in hormone-driven cancer models. Xanthene and pyrene derivatives have been widely reported to exhibit diverse biological activities, including anticancer effects, which provided a rationale for selecting these chemical scaffolds for hybrid design [11,12]. The novelty of our work lies in testing 11 compounds that were previously evaluated for antimicrobial activity [14] for the first time in the context of cancer, alongside 11 entirely new hybrids never before assessed. This approach allows us to explore the antitumor potential of both previously known and novel hybrids, while investigating their mechanisms of action and possible simultaneous targeting of signaling and cell viability pathways.

## 2. Materials and Methods

### 2.1. Hybrid Compounds

The hybrid compounds used in this study consisted of xanthene-dihydropyrimidinone hybrids. The xanthene-dihydropyrimidinone hybrids were synthesized by connecting both pharmacophore units through a triazole heterocyclic bridge, formed via a copper-catalyzed alkyne–azide cycloaddition reaction (CuAAC). In this approach, the alkyne moiety was anchored to the xanthene scaffold, while the azide group was introduced into the dihydropyrimidinone (DHPM) derivatives. These molecules were selected based on previous work and corresponded to the compounds ensuring consistency with the previously characterized chemical structures and antimicrobial activities [14]. Chemistry structure (Figure 1) and molecular weight of 22 molecules are available in Appendix A. The preparation of the hybrid compounds can be found in Appendix A, where the molecular hybridization points are also indicated. Detailed spectral characterization of each compound, including ^1^H NMR, ^13^C NMR, and high-resolution mass spectrometry (HRMS) data, is provided in the Characterization of Hybrid Compounds section at the end of the Appendix A.

### 2.2. Cell Lines and Culture Conditions

Breast cancer cell lines MDA-MB-231 (triple-negative), T-47D (Luminal A), BT-474 (Luminal B), and SKBR-3 (HER2-positive); ovarian cancer cell lines SKOV-3, A2780, OVCAR-3, and PA-1; and prostate cancer cell lines PC-3 and LNCaP were used in this study. Normal human fibroblasts (HFF-1) were employed as a control. All cell lines were commercially obtained from the European Collection of Cell Cultures (ECACC, Salisbury, UK) and the American Type Culture Collection (ATCC, Manassas, VA, USA) and deposited in the Cell Bank of the Barretos Cancer Hospital. All cell lines tested negative for mycoplasma contamination. In general, cells were cultured in DMEM or RPMI medium supplemented with 10% fetal bovine serum (FBS, Gibco, Waltham, MA, USA) and 1% penicillin/streptomycin (P/S), in culture flasks, and incubated at 37 °C in a humidified atmosphere containing 5% CO_2_ and 90% humidity until reaching confluence. Once confluent, cells were trypsinized, seeded, and maintained under the same conditions for biological characterization and therapeutic response assays.

### 2.3. Cytotoxicity Assessment and Screening of Hybrid Molecules

The standard protocol for candidate molecule screening developed by the National Cancer Institute (NCI) was employed in the primary analysis, referred to as the Cell One-Dose Screen [15]. Cell lines were treated to fixed concentration (10 mM) of each hybrid’s molecules for 72 h. The hybrid compounds that reduced cell viability in less than 50% of tumor cells and maintained viability above 50% in non-tumoral cells were selected for the next stage. Subsequently, the most promising molecules were subjected to a second screening stage, termed the Cell Five-Dose Screen. For this purpose, 1 × 10^4^ cells were seeded in 96-well plates treated with increasing concentrations (0, 2, 5, 10, 15, 20, 25, and 35 μM) of each along with the selected treatments for 72 h. After treatment, all cell lines were fixed with trichloroacetic acid 4% and were then stained with SRB solution 0.4%, and excess unbound dye was removed through successive washes with 1% acetic acid. Finally, the protein-bound dye was solubilized in a basic Tris-containing solution, and absorbance was measured at 595 nm using microplate reader Varioskan (Thermo Scientific, Waltham, MA, USA). The IC_50_ values were determined by nonlinear regression analysis using GraphPad Prism, version 9.0.1 based on dose–response curves. The selectivity index (SI) was calculated as the ratio of IC_50_ in the non-tumoral HFF-1 fibroblast line to IC_50_ in tumor cells (SI = IC_50_ HFF-1/IC_50_ tumor), with values greater than 2 indicating preferential cytotoxicity toward tumor cells.

### 2.4. Cell Cycle and Cell Death Analysis

To evaluate cell death by apoptosis, tumor cells were treated with the most promising hybrid compounds, selected in the previous screening steps, at concentrations equivalent to their respective IC_50_ values. Cells were then stained with Annexin V (for apoptosis detection) and propidium iodide (PI, for cell viability assessment) using the Annexin V-FITC Apoptosis Detection Kit I (BD Biosciences, Milpitas, CA, USA), according to the manufacturer’s instructions. For cell cycle analysis, cells were treated with the selected compounds at their IC_50_ concentrations, stained with propidium iodide (Cycle Test Plus Kit, BD Biosciences), and analyzed on a BD Accuri C6 flow cytometer (BD Biosciences, San Jose, CA, USA) using the manufacturer’s protocol.

### 2.5. Phospho-Kinase Array

The T47-D cell line (sensitive cell line) was seeded in 6-well plates and treated with the selected compound for 24 h. Cell lysates were then prepared according to the manufacturer’s instructions using the Lysis Buffer 6 provided in the commercial kit Proteome Profiler Human Phospho-Kinase Array Kit (Catalog # ARY003C, R&D Systems, Minneapolis, MN, USA). Following protein quantification by the Bradford method, 700 µg of total protein from each condition were incubated with the pre-blocked membranes overnight, allowing the specific binding of phosphorylated proteins to the immobilized antibodies. Detection was performed by chemiluminescence using HRP-conjugated secondary antibodies and visualization with an ECL-compatible imaging system. Densitometric analysis of the spots was carried out using ImageJ 1.54p software (NIH, Bethesda, MA, USA), and the values were normalized to the positive and negative controls, as recommended by the manufacturer.

### 2.6. Assessment of Proliferative Pathway Inhibition by Western Blotting

After treatment of tumor cells with the most promising hybrid molecules (SJ028, SJ064, SJ078) of each type of tumor, protein extracts were prepared. Cells were cultured in T25 flasks and, upon reaching approximately 90% confluence, were treated for 24 and 48 h. Total protein content from each cell line was quantified using the Bradford method, and proteins were subsequently separated by denaturing polyacrylamide gel electrophoresis (SDS-PAGE). Following separation, proteins were transferred to nitrocellulose membranes and incubated overnight with primary antibodies (1:1000) against ERK1/2 (Cat n° 4370), total AKT (Cat n° 4691), phospho-AKT (Cat n° 9271), p21 (Cat n° 2947), phospho-Stat3 (Cat n° 9131), phospho-H2AX (Cat n° 9718), PI3K (Cat n° 4228), and phospho-mTOR (Cat n° 2971). Membranes were then washed and incubated with anti-rabbit or anti-mouse secondary antibodies (1:5000) for 2 h at room temperature. Protein bands were detected using the Signal Fire ECL system (Cell Signaling, Danvers, MA, USA), and images were acquired with the Image Quant LAS 4000mini documentation platform (GE Health Sciences, Chicago, IL, USA).

### 2.7. AKT Silencing by esiRNA

Cell lines were seeded in 6-well plates at a density of 3.5 × 10^5^ cells/mL in DMEM supplemented with 10% fetal bovine serum and incubated overnight to allow adherence. Subsequently, cells were transfected with 28 pmol of (Pan) esiRNAs specific for AKT1, AKT2, and AKT3 isoforms (esiAKT1/2/3; Sigma-Aldrich, Saint Louis, MO, USA) to achieve simultaneous silencing of all three isoforms. A luciferase-targeting esiRNA (esiLuciferase; Sigma-Aldrich) was used as a negative control. Transfection was performed using Lipofectamine™ 3000 (Thermo Fisher Scientific, L3000008) in reduced-serum medium (Opti-MEM, Gibco, 11058021) without antibiotics for 6 h. After transfection, cells were seeded into black, clear-bottom 96-well plates and maintained for up to 72 h. Cell viability was monitored in real-time using the RealTime-Glo™ MT Cell Viability Assay (Promega, Madison, WI, USA, G9711) by adding diluted substrate and enzyme (1:500) to the culture medium, with readings taken every 24 h using a Varioskan™ LUX plate reader (Thermo Fisher Scientific, Waltham, MA, USA). Gene silencing efficiency was confirmed by Western Blot analysis 48 h post-transfection using a pan-AKT primary antibody (Akt (pan) (C67E7) Rabbit mAb #4691, Cell Signaling) followed by appropriate secondary antibodies. Protein bands were visualized using enhanced chemiluminescence (ECL) detection.

### 2.8. AKT Inhibition by Pharmacological Agent

To evaluate combinatorial effects between the pharmacological AKT inhibitor MK2206 (Selleck, Houston TX, USA, S1078) and the hybrid molecules, a dilution matrix containing four concentrations of the hybrid molecules (0, IC_25_, IC_50_, and IC_75_) and MK2206 (0, IC_25_, IC_50_, and IC_75_) was used. Following exposure to the combinations, absorbance was measured by SRB assay using a microplate reader. Raw data were analyzed using the web application SynergyFinder 3.0—a web application for interactive analysis and visualization of drug combination screening data. Available online: https://synergyfinder.fimm.fi/ (accessed on 16 September 2024), and combinatorial effects were calculated according to Bliss, ZIP, and Loewe criteria to identify synergistic, antagonistic, or additive interactions.

### 2.9. Three-Dimensional Culture Model (Spheroids)

To establish optimized 3D models, cells were initially cultured under standard supplemented conditions in culture flasks at 37 °C, 5% CO_2_, and 90% humidity until confluence. Cells were then seeded into 96-well plates pre-coated with adhesion medium (RPMI or DMEM, depending on the cell line). Nunclon™ Sphera™ 96-well U-bottom plates (Thermo Scientific™, 174925) were used to promote uniform and stable spheroid formation. Cells were incubated for 4 days under the same conditions until spheroid formation was visually confirmed. Baseline images were captured using an inverted microscope. Seeding densities were adapted according to the cell line: 1 × 10^4^ cells/well for breast lines (e.g., T-47D), 2 × 10^4^ cells/well for prostate lines (e.g., LNCaP and PC-3), and 20 × 10^4^ cells/well for ovarian lines (e.g., OVCAR-3, A2780, and SKOV-3) with 0.25% methylcellulose added to enhance spheroid formation. After baseline imaging, spheroids were treated with selected hybrid molecules. Morphological changes induced by treatments were monitored by photographic records at 24, 48, and 72 h intervals, and dynamic cellular responses were assessed via caspase 3/7 activity, morphological alterations, and cell viability using the CellTiter-Glo^®^ 3D Cell Viability Assay (Promega, G9681) and LIVE/DEAD™ Viability/Cytotoxicity Kit for Mammalian Cells (Thermo Fisher Scientific, L3224)

### 2.10. Determination of Combination Index (C.I.) for Drugs

Therapeutic combinations were evaluated using a dilution matrix containing four concentrations of hybrid molecules (0, IC_25_, IC_50_, and IC_75_) and hormonal therapies (tamoxifen, fulvestrant, or letrozole) at 0, IC_25_, IC_50_, and IC_75_ for each hormone therapy in the selected cell lines. Tamoxifen acts as a selective estrogen receptor modulator (SERM), blocking estrogen binding and inhibiting ER-mediated transcriptional activity. Fulvestrant is a selective estrogen receptor degrader (SERD) that promotes receptor degradation and complete suppression of estrogen signaling. In contrast, letrozole is a nonsteroidal aromatase inhibitor that reduces estrogen synthesis by inhibiting the aromatase enzyme. Due to their distinct mechanisms of action, these drugs are widely used in hormone-dependent cancers and may also modulate estrogen signaling in ovarian and prostate models expressing ER [2]. Following combination exposure, absorbance was measured at 595 nm by SRB assay using a microplate reader. Raw data were analyzed with SynergyFinder 3.0—a web application for interactive analysis and visualization of drug combination screening data. Available online: https://synergyfinder.fimm.fi/ (accessed on 16 September 2024), and combinatorial effects were calculated using Bliss, ZIP, and Loewe criteria to identify synergistic, antagonistic, or additive interactions.

### 2.11. Acquired Antihormone Resistance Models

To generate hormone therapy-resistant cell lines, T-47D cells were subjected to intermittent treatment with tamoxifen, fulvestrant, and letrozole, whereas OVCAR-3 and LNCaP lines were exposed exclusively to tamoxifen. The protocol involved cycles of 3 days of treatment followed by 3 days of recovery in a complete medium containing 10% fetal bovine serum, according to previously established acquired resistance models [16]. Initial exposure for T-47D cells was performed with 100 nM 4-hydroxytamoxifen (the active metabolite of tamoxifen), 100 nM fulvestrant, and 100 nM letrozole, with gradual doubling of concentrations in each cycle according to cellular tolerance. The same approach was applied to OVCAR-3 and LNCaP starting at 100 nM 4-hydroxytamoxifen. Cells were maintained under this regimen until the IC_50_ reached at least 35 µM, indicating acquired resistance. The resistance induction phase lasted 66 days. Cell sensitivity was assessed every four weeks via SRB viability assays to monitor resistance development. Adaptation to letrozole and fulvestrant followed sublethal concentration protocols previously described for progressive resistance in luminal breast cancer models. Upon confirmation of resistance, cell lines were maintained at elevated concentrations to stabilize the resistant phenotype. For the determination of the IC_50_ values of resistant cells, tamoxifen, fulvestrant, and letrozole were added to the cultures at the same concentrations employed in the minimum Five-Dose Screening (0, 2, 5, 10, 15, 20, 25, and 35 μM). To maintain the resistant phenotype, cells were continuously cultured in the presence of 51.2 μM of the respective hormone therapy, corresponding to the highest concentration previously used during the induction of resistance.

### 2.12. Molecular Docking

To perform molecular docking analyses, we retrieved the crystallographic structure of human AKT1 in complex with an allosteric inhibitor (PDB ID: 3O96; resolution: 2.70 Å), specifically AKT inhibitor VIII (AVIII), from the Protein Data Bank. The co-crystallized ligand (AVIII) was extracted and saved in PDB format to serve as a control for redocking validation. The SMILES sequence of the test compounds SJ028, SJ064, and SJ078 were converted to three-dimensional structures in PDB format using the OpenBabel library. Receptor and ligand preparations were carried out using MGLTools 1.5.7 scripts: prepare_receptor4.py for the protein and prepare_ligand4.py for the ligands. These scripts were employed to add polar hydrogens, compute Gasteiger partial charges, and detect rotatable bonds for flexible docking. The docking grid was centered based on the center of mass of the co-crystallized ligand, with a grid box size of 24 × 24 × 24 Å. Docking simulations were performed using AutoDock Vina version 1.2.0, with an exhaustiveness value of 20 and Vina’s default scoring function, which includes non-directional hydrogen-bond and hydrophobic terms, steric components (two attractive Gaussian functions and a repulsion term), and a conformational entropy penalty, as described by Trott and Olson (2010) and further extended in Vina 1.2.0 [17]. In our docking protocol, the AKT1 receptor was treated as rigid, while ligands were considered flexible, allowing torsional freedom around identified rotatable bonds. The search space was defined based on the crystallographic ligand coordinates to ensure accurate targeting of the known binding site.

For evaluation, the top-ranked poses of each ligand were selected based on the lowest predicted binding affinity. To analyze molecular interactions, we utilized BINding ANAlyzer (BINANA) python script, which detects hydrogen bonds, hydrophobic contacts, salt bridges, and other key interactions at the binding interface based on the default cut-off interaction distances. Additionally, BIOVIA Discovery Studio Visualizer [18] was employed to perform a consensus-based inspection and graphical visualization of protein-ligand interactions.

### 2.13. Statistical Analysis

All experimental data were analyzed using GraphPad Prism, version 9.0.1. Data are presented as mean ± standard deviation (SD) from at least two independent biological replicates. Comparisons between each treatment and the DMSO control were performed using one-way ANOVA followed by Dunnett’s post hoc test to correct for multiple comparisons. Differences were considered statistically significant when *p* < 0.05. Statistically significant differences are indicated in the figures with **.

## 3. Results

### 3.1. Hybrid Dihydropyrimidinones Reduce Viability in Hormone-Driven Cancer in 2D and 3D Models

#### 3.1.1. Screening of Xanthene-Dihydropyrimidinone Hybrid Compounds

In the initial screening, breast (MDA-MB-231, T-47D, BT-474, SKBR-3), prostate (PC-3, LNCaP), and ovarian (OVCAR-3, SKOV-3, A2780) cancer cell lines, as well as the non-tumoral fibroblast line HFF-1, were treated with were treated with 11 xanthene-dihydropyrimidinone and 11 pyran-dihydropyrimidinone hybrid compounds at 10 μM for 72 h, with DMSO as the vehicle control. Cell viability varied widely across tumor types, ranging from 3 to 107% in prostate, 20 to 153% in breast, and 4 to 108% in ovarian cancer lines, whereas none of the compounds reduced HFF-1 viability by more than 50%, indicating favorable initial selectivity towards cancer cells (Figure 2I,J).

During the one-dose screening, differential sensitivity to hybrid molecules was observed across tumor cell lines. MDA-MB-231, T-47D, SKBR-3, LNCaP, PC-3, SKOV-3, and A2780 responded to a subset of compounds, whereas BT-474 showed no sensitivity. Notably, OVCAR-3 exhibited broad sensitivity to all 22 hybrids tested (Figure 2A–H,K–T). Within the prostate cancer panel, LNCaP was the most sensitive model, with 12 compounds reducing viability by >50%, compared with five in PC-3. In breast cancer, T-47D responded to 16 molecules, MDA-MB-231 to 14, and SKBR-3 to 4, while BT-474 was largely resistant. In ovarian cancer, OVCAR-3 showed strong sensitivity to all 22 molecules, whereas SKOV-3 responded to 15.

All molecules that passed the one-dose screening were subsequently evaluated in the five-dose assay, but only those exhibiting the lowest IC_50_ values for each tumor type are presented in Table 1: SJ028 and SJ064 for breast cancer, SJ028 and SJ078 for prostate cancer, and SJ028 for ovarian cancer. IC_50_ values for the selected molecules ranged from 3.44 ± 1.42 to 36.62 ± 1.34 μM (Table 1). The selectivity index (SI) was calculated as the ratio of IC_50_ in HFF-1 to IC_50_ in tumor cells (SI = IC_50_ of HFF-1/IC_50_ of tumor cell), with SI > 2 considered indicative of tumor-selective activity. Using this criterion, the most responsive hormone-dependent models were T-47D (SJ028 and SJ064), LNCaP (SJ028 and SJ078), and OVCAR-3 (SJ028), demonstrating that SJ028 consistently exhibits potent and selective antitumor activity across tumor types. All SI values for the molecules tested in the minimum five-dose screening across all cell lines are provided in Appendix A.

The physicochemical properties of the hybrid compounds SJ028, SJ064, and SJ078 were evaluated according to Lipinski’s Rule of Five. All compounds presented two violations of the rule, mainly due to their high molecular weight and the number of hydrogen-bond acceptors. These deviations indicate that, although the compounds exceed some conventional thresholds for oral bioavailability, they remain promising drug candidates considering their potent biological activity. Detailed values for each parameter are provided in Appendix A.

Morphological and viability analyses of T-47D (treated with SJ028 and SJ064), LNCaP (treated with SJ028 and SJ078), and OVCAR-3 (treated with SJ028) spheroids over a 72 h period revealed significant alterations induced by hybrid molecules. Representative images (Figure 3) showed progressive changes in spheroid perimeter, particularly in LNCaP (Figure 3E) and OVCAR-3 (Figure 3I), indicating growth inhibition. Live/dead staining (Figure 3B,F,J) highlighted increased cell death in treated spheroids, consistent with the positive control 5-FU, while DMSO-treated spheroids remained largely viable. Quantitative assessment (Figure 3C,G,K) confirmed significant reductions in cell viability. Western Blot analysis (Figure 3D,H,L) demonstrated upregulation of γH2AX phosphorylated in treated spheroids, indicating DNA damage, and supporting the cytotoxic effects of the selected hybrid compounds in 3D tumor models.

#### 3.1.2. Hybrid Molecules Induce Cell Cycle Arrest and Apoptosis in Hormone-Driven Cancer Cells

For mechanistic investigations, the most sensitive cell line of each tumor type: T-47D (breast), LNCaP (prostate), and OVCAR-3 (ovary), were selected for flow cytometry analysis. In T-47D cells, treatment with compounds SJ028 and SJ064 increased both early and late apoptosis after 24 h and induced cell cycle arrest in S and G2/M phases at 48 h (Figure 4E). Similar effects were observed in LNCaP cells treated with SJ028 and SJ078 (Figure 4K), and in OVCAR-3 cells exposed to SJ028 (Figure 4P).

Cell cycle analysis revealed that hybrid compounds induced S-phase arrest across all cell lines, accompanied by G2 arrest in T-47D cells, blocking S–G2/M transitions after 48 h of treatment (Figure 4). In T-47D cells, treatment with SJ028 reduced the G1 population (from 71.1% in DMSO to 9.5%), increased the S-phase fraction (from 18.7% to 39.5%), and elevated the G2/M population (from 9.3% to 44.4%). SJ064 similarly decreased G1 (26.1%), enhanced S-phase (45.7%), and moderately increased G2/M (26.7%). In contrast, 5-FU caused a pronounced decrease in G1 (to 35.5%), a strong accumulation in S-phase (64.9%), and a marked reduction in G2/M (0.99%). In LNCaP cells, SJ028 markedly shifted the cycle profile, reducing the G1 population from 67.5% to 18.7%, strongly increasing S-phase from 14.0% to 79.6%, and decreasing G2/M from 15.8% to 1.7%. SJ078 induced a similar but less pronounced effect, lowering G1 to ~33.0%, elevating S-phase to ~67.5%, and nearly abolishing G2/M (<2%). In OVCAR-3 cells, SJ028 produced a dramatic G1 decrease (34.1% to 3.8%), robust S-phase accumulation (23.4% to 75.4%), and reduced G2/M (19.9% to 11.4%). Comparable to this profile, 5-FU promoted strong S-phase accumulation (74.7%), with partial G1 preservation (29.5%) and near-complete G2/M loss (1.4%). In all conditions, p21 upregulation corroborated cell cycle arrest at 48 h (Figure 4F,L,Q).

To further confirm these findings in more complex cellular models, 3D spheroid models were employed to evaluate caspase 3/7 activity, morphology, and cell viability using live/dead assays. In T-47D spheroids, hybrid compounds SJ028 and SJ064 markedly reduced cell viability and increased apoptosis and necrosis (Figure 5A–G). Similar results were obtained in LNCaP spheroids treated with SJ028 and SJ078, and OVCAR-3 spheroids treated with SJ028 (Figure 5H–T). These observations confirm the cytotoxic efficacy of hybrid compounds in both 2D and 3D models, demonstrating their potential for robust antitumor activity in tridimensional tumors models.

### 3.2. AKT Signaling Is a Molecular Target of Xanthene-Dihydropyrimidinone Hybrid Compounds

To identify potential molecular targets of the xanthene-dihydropyrimidinone hybrids, a phospho-kinase array was performed in T-47D cells (sensitive) treated with SJ028. Increased phosphorylation levels were observed for GSK, CREB, Src, FAK, and c-JUN, whereas β-catenin and pRAS40 were decreased after treatment (Figure 6A,B). These alterations suggest inhibition of proliferation and survival associated pathways connected to PI3K/AKT signaling. Enhanced phospho-γH2AX phosphorylated expression and reduced pRAS40 levels indicated DNA damage and functional AKT inhibition, even in the absence of changes in PI3K and mTOR phosphorylated protein (Figure 6D–F) confirm that treatment with SJ028 and SJ064 molecules inhibited AKT phosphorylation after 72 h of treatment in the T47-D tumor cell line (Figure 6G–I). SJ078 treatment markedly reduced AKT levels in both LNCaP and OVCAR-3 cells after 48 h, while total PI3K and mTOR remained unaffected (Figure 6D,G), indicating a selective modulation of the AKT-mediated pathway.

Molecular docking simulations revealed that SJ028 can directly interact with the AKT1 allosteric site with favorable binding energy (Figure 6J; Appendix A), supporting the hypothesis of allosteric inhibition of AKT1 isoform. The interaction analysis for the control ligand, AVIII, revealed several key stabilizing contacts (Figure 6J). Its binding mode is characterized by a crucial π-cation interaction with ARG273 and a π-π stacked interaction with TRP80. Additional stability is conferred by π-alkyl interactions with LEU264 and LYS268, and a conventional hydrogen bond with SER205.

The test compounds, while sharing some features with AVIII, displayed distinct interaction profiles (Figure 6J). A conserved feature across SJ028, SJ064, and SJ078 was the critical π-π stacked interaction with TRP80. However, a notable difference was the absence of the π-cation interaction with ARG273 residue. Instead, these compounds appear to compensate for this loss through an extensive network of hydrogen bonds. For example, SJ028, SJ064, and SJ078 all formed multiple hydrogen bonds with key polar residues, including ASN54, THR82, and ASP274, effectively anchoring them within the pocket. Furthermore, all three compounds established a π-sigma interaction with TYR272, contributing additional stability (Figure 6J). In summary, while the test compounds lack the π-cation interaction observed for the control, they achieve high predicted affinity through a combination of the conserved π-π stacking with TRP80 and a dominant network of hydrogen bonds. This suggests an alternative but potent binding mechanism within the AKT1 allosteric pocket, highlighting their promise as novel inhibitors.

To validate the docking protocol, the co-crystallized inhibitor Akt inhibitor VIII (AVIII) was redocked into the allosteric pocket of AKT1. The procedure yielded a top-ranked pose with a binding free energy of −14.3 kcal/mol and a root-mean-square deviation (RMSD) of 0.18 Å from the crystallographic ligand coordinates (Appendix A), thereby confirming the protocol’s reliability in reproducing the experimental binding conformation. The compounds SJ028, SJ064, and SJ078 exhibited strong predicted binding affinities of −12.3, −12.5, and −11.1 kcal/mol, respectively. Although slightly lower than the control, these values indicate a high binding potential for the AKT1 allosteric site (Appendix A).

In addition, we performed validation experiments demonstrating that the cytotoxic activity of the hybrid compounds depends on AKT. Both pan-allosteric AKT inhibition using MK2206 and AKT gene silencing via esiRNA impaired the activity of the hybrids (Figure 6C,D). Combination assays with MK2206 consistently revealed antagonistic effects across all tumor models. In T-47D cells, SJ028 + MK2206 showed strong antagonism (ZIP–37.56, Bliss–36.04, Loewe–27.26), while SJ064 + MK2206 produced a milder but still antagonistic effect (ZIP–12.24, Bliss–12.65, Loewe–8.07). In LNCaP cells, SJ028 + MK2206 exhibited marked antagonism (ZIP–46.84, Bliss–45.96, Loewe–30.83), as did SJ078 + MK2206 (ZIP–45.58, Bliss–45.09, Loewe–34.04). Similarly, in OVCAR-3 cells, SJ028 + MK2206 showed antagonistic interactions (ZIP–17.41, Bliss–16.57, Loewe–12.82) (Appendix A). Across all models, these combinations were classified as antagonistic according to all three reference models. Importantly, esiRNA silencing of AKT isoforms abolished the cytotoxic effects of the hybrid compounds. Figure 7G–I shows that in AKT-silenced cells, SJ028 and SJ064 no longer reduced cell viability, whereas control cells (positive AKT expression) exhibited a strong viability reduction (~70%). These results confirm that AKT is a critical functional target for the cytotoxic activity of the hybrid compounds.

### 3.3. Hormone Therapies Synergize with Xanthene-Dihydropyrimidinone Hybrid Compounds in Hormone-Driven Cancer Lines

The IC_50_ values for hormone therapies (tamoxifen, fulvestrant, and letrozole) were determined in T-47D, LNCaP, and OVCAR-3 cells (Figure 8A–C; Appendix A) Tamoxifen was effective in all three lines, fulvestrant only in T-47D and OVCAR-3, and letrozole only in T-47D. These differences reflect the distinct hormonal receptor profiles of each cell line. Combined treatments between hybrid molecules SJ028, SJ064 and SJ078 and hormone therapies (tamoxifen, fulvestrant and letrozole) were analyzed using ZIP, Bliss, and Loewe synergy models. The OVCAR-3 cell line showed sensitivity only to tamoxifen (IC_50_ = 0.0174 µM) and fulvestrant (IC_50_ = 3.047 µM), whereas LNCaP cells were sensitive exclusively to tamoxifen (IC_50_ = 0.2838 µM), with fulvestrant and letrozole being non-effective. T-47D cells exhibited sensitivity to all three hormonal therapies, with IC_50_ values of 0.3035 µM for tamoxifen, 0.7685 µM for fulvestrant, and 1.635 µM for letrozole. These data indicate cell line-specific differences in response to the evaluated hormonal therapies (Appendix A). Synergistic effects were observed in T-47D treated with SJ028 (ZIP = 64.73, Bliss = 64.58, Loewe = 44.71) and SJ064 (ZIP = 35.75, Bliss = 35.70, Loewe = 5.63), as well as in OVCAR-3 treated with SJ028 (ZIP = 25.68, Bliss = 24.30, Loewe = 8.44), all of which exhibit higher estrogen receptor expression. In contrast, LNCaP cells displayed predominantly additive interactions with SJ028 (ZIP = −2.10, Bliss = −3.70, Loewe = −23.81) and SJ078 (ZIP = −3.84, Bliss = −6.28, Loewe = −18.02) (Figure 7E; Appendix A).

### 3.4. Xanthene-Dihydropyrimidinone Hybrid Compounds Are Effective in Hormone Therapy-Resistant Cancer Models

To assess the activity of hybrid molecules in hormone-resistant settings, resistance was induced by intermittent exposure to hormonal agents: T-47D cells to tamoxifen, fulvestrant, and letrozole; OVCAR-3 cells to tamoxifen and fulvestrant; and LNCaP cells to tamoxifen. Resistance was confirmed through functional viability assays and biomarker analysis (PARP cleavage and γH2AX phosphorylated) (Figure 9E–G,I,J), along with morphological changes in resistant cultures (Figure 9D,H).

Importantly, hybrid compounds SJ028, SJ064, and SJ078 retained IC_50_ values comparable to those observed in parental lines (Figure 9B,C; Appendix A), indicating preserved cytotoxic efficacy despite acquired hormone therapy resistance. Among the resistant models, SJ028 consistently displayed low IC_50_ values, including 4.36 µM in tamoxifen-resistant T-47D, 4.37 µM in fulvestrant-resistant T-47D, and 4.51 µM in letrozole-resistant T-47D cells. In LNCaP tamoxifen-resistant cells, SJ028 was notably more efficient (1.60 µM) compared to SJ078 (16.18 µM). In OVCAR-3 resistant models, SJ028 again maintained lower IC_50_ values (9.72 µM), with SJ064 showing a slightly higher efficiency in the fulvestrant-resistant setting (11.39 µM).

## 4. Discussion

The present study highlights the potential of xanthene-dihydropyrimidinone hybrids, originally investigated for antibacterial and antibiofilm properties [14], as promising candidates for oncological evaluation. Xanthene scaffolds have consistently been associated with antineoplastic activity in preclinical models, exhibiting cytotoxic, antiproliferative, and pro-apoptotic effects across breast, prostate, and ovarian cancer systems [19,20,21]. While pyrene derivatives are particularly known for inducing apoptosis and cell cycle arrest [22]. Xanthene-based compounds have demonstrated micromolar cytotoxicity and inhibition of tumor cell migration. These preliminary findings support the rationale for investigating the 22 newly synthesized hybrids in this study, which had not been previously assessed for their antineoplastic potential. Given the clinical prevalence of estrogen- and androgen-driven tumors, evaluating these compounds in hormone-dependent cancer cell lines is particularly relevant [23]. This work not only assessed the antineoplastic activity of xanthene-dihydropyrimidinone hybrids and their interaction with hormone receptor–mediated pathways, but also established a platform for identifying candidate molecules able of overcoming resistance in hormone-dependent cancers. Although all hybrid compounds presented two Lipinski’s rule violations, this does not preclude their potential as drug candidates. The rule of five, proposed by Lipinski in 1997, is an empirical guideline estimating the likelihood of good oral bioavailability for small molecules [24]. However, many approved drugs violate one or more of these parameters [25]. Particularly, complex or hybrid scaffolds, such as dihydropyrimidinone and xanthene derivatives, often exceed molecular weight or hydrogen-bond thresholds while maintaining potent biological activity [26,27]. Therefore, these deviations do not necessarily indicate poor pharmacokinetic performance but rather suggest that formulation optimization or alternative administration routes could be explored without compromising pharmacological relevance [28].

The hybrid molecules exhibited consistent cytotoxicity in both 2D cultures and 3D spheroid models, reinforcing their translational potential [29]. The selectivity indices for SJ028, SJ064, and SJ078 were favorable for oncological drug development (IC_50_ < 35 µM) [30]. These results align with previous reports showing that xanthene derivatives modulate cell cycle checkpoints, inducing S- or G_2_/M-phase arrest in various tumor types. For instance, cowaxanthone 5 from Garcinia cowa induced S-phase arrest in HeLa cells; γ-mangostin blocked S-phase progression in colorectal cancer cells [31], and xanthatin induced G_2_/M arrest in A549 lung cancer cells via p53 activation and NF-κB inhibition [32]. Consistently, SJ028, SJ064, and SJ078 blocked the cell cycle in S and G_2_/M phases, likely mediated, at least in part, by the cyclin-dependent kinase inhibitor p21 (CDKN1A), a canonical mediator of DNA damage–induced cell cycle arrest regulated by p53 [33].

The tumor lines selected, while sharing hormone-dependent mechanisms, present distinct mutational contexts that may influence their responses to hybrid molecules treatments. For example, T-47D cells harbor a *TP53* L194F missense mutation, impairing p53 DNA-binding and its regulation of cell cycle arrest and apoptosis [34]; OVCAR-3 cells carry an *TP53* R248Q mutation, which confers oncogenic properties and disrupts canonical tumor suppressor functions [35], while LNCaP cells present *TP53* M160I mutation compromising p53 transcriptional activity [36]. These alterations create a context in which p53-dependent pathways are partially or fully impaired, providing an ideal setting to evaluate compounds acting independently of p53 or aiming to restore downstream signaling. This context also explains variations in sensitivity to the xanthene-dihydropyrimidinone hybrid and highlights the potential of targeting alternative survival pathways, such as AKT signaling. Differential sensitivity across T-47D, LNCaP, and OVCAR-3 cells reflected their molecular backgrounds. T-47D cells, harboring *PIK3CA* H1047R, were highly sensitive; LNCaP cells, with *PTEN* loss and elevated AKT2/3, were relatively resistant; OVCAR-3 cells, with constitutive AKT activation, showed reduced responsiveness. These trends are consistent with previous findings that xanthone derivatives, including gambogic acid, induce apoptosis via AKT inhibition and that efficacy depends on the tumor molecular profile [37,38,39,40]. Accumulation of γH2AX further supports DNA damage and impaired repair via AKT and downstream effectors (CHK1, BRCA1, MDM2) as a key cytotoxic mechanism.

Mechanistically, the xanthene-dihydropyrimidinone inhibited AKT kinase, a central regulator of cellular survival. AKT inhibition decreased phosphorylation of anti-apoptotic proteins such as BAD and MDM2, promoting activation of caspase-9 and downstream caspases 3 and 7, ultimately leading to apoptosis [41,42]. The increased in GSK-3 levels observed in our study, may represent a compensatory mechanism for AKT-mediated activation, since xanthene-dihydropyrimidinone hybrid appear to have a direct inhibitory effect on AKT. In addition, we observed a reduction in β-catenin levels, which may suggest an effect of AKT/β-catenin axis suppression, as reported in other studies [13,43]. Conversely, the increase in CREB levels after treatment with xanthene-dihydropyrimidinone hybrid may have triggered tumor plasticity mechanisms [44]. In summary, we demonstrated that SJ028, SJ064, and SJ078 reduced AKT phosphorylation across T-47D, LNCaP, and OVCAR-3 cells, accompanied by decreased levels of downstream effectors, including pRAS40, an AKT substrate regulating mTORC1 signaling [37]. Activation of stress-related proteins (c-JUN) and γH2AX accumulation indicated DNA damage and genotoxic potential of xanthene-dihydropyrimidinone hybrid.

Molecular docking confirmed that both SJ028, SJ064, and SJ078 hybrids stably bind the AKT1 allosteric site, forming conserved π-π stacking with TRP80 and hydrogen bonds with ASN54, THR82, and ASP274, providing a structural rationale for functional inhibition. Interestingly, we conducted therapeutic combinations with the allosteric AKT inhibitor (MK2206) combinate with the molecules SJ028, SJ064, and SJ078 and found clear antagonistic action against the molecules evaluated. This fact can be explained by the strong binding of MK2206 to the phosphorylation residues T308 and S473 [38] where the xanthene-dihydropyrimidinone hybrid compounds also have binding affinity. When all AKT isoforms were knocked down, we observed a clear reduction in sensitivity to SJ028, SJ064, and SJ078 molecules, confirming AKT selectivity inhibition.

Both hybrids’ molecules SJ028, SJ064, and SJ078, were able to promote cycle arrest and trigger apoptosis mechanisms in the tumor cell lines tested in our study. The effect of hybrid molecules can also be detected in three-dimensional models, which showed caspase 3/7 activation and cell death. Studies demonstrate that AKT inhibition primarily induces cell cycle arrest, particularly at the G1/S and G2/M phases, by increasing the expression of cyclin-dependent kinase inhibitors (CDKIs) like p27, p21 and decreasing the activity of cyclin-dependent kinases (CDKs) that promote cell cycle progression [39,40].

Besides their considerable cytotoxic effect as a single agent, the hybrids acted synergistically with hormone therapies in cell lines that were already sensitive to hormone therapies. In T-47D cells, SJ028 and SJ064 synergized with tamoxifen, fulvestrant, and letrozole; in OVCAR-3 cells, synergy occurred with tamoxifen and fulvestrant; in LNCaP cells, interactions were primarily additive. A recent study demonstrated that capivasertib (anti-AKT)-fulvestrant therapy resulted in significantly longer progression-free survival than fulvestrant alone among patients with advanced hormone receptor-positive breast cancer [45]. Our results suggest hybrids can enhance standard hormone therapy efficacy, particularly in tumors with high hormone receptor expression.

In the secondary or acquired resistance hormone type, the tumor cells initially respond to the hormone therapy but then develops resistance over time, often after long-term exposure to the treatment [46]. In our study, resistant cell lines were generated through intermittent exposure to hormonal agents. Acquired resistance was validated in T-47D cells to tamoxifen, fulvestrant, and letrozole; in OVCAR-3 cells to tamoxifen and fulvestrant; and in LNCaP cells to tamoxifen. Crucially, resistant cells remained sensitive to SJ028, SJ064, and SJ078, with IC_50_ values comparable to parental lines. However, SJ028 consistently showed the lowest IC_50_ value, highlighting its potential as a hybrid molecule capable of acting on cells that have acquired resistance to hormonal therapies. The activation and/or overexpression of AKT is one of the mechanisms already reported in hormone therapy resistance phenotypes in resistant breast [47] and prostate tumors [48]. This fact may explain the unchanged sensitivity to SJ028, SJ064, and SJ078 molecules in the resistant models evaluated.

Collectively, these results integrate functional, molecular, and structural evidence to establish that xanthene-dihydropyrimidinone hybrids target AKT signaling to induce DNA damage, cell cycle arrest, and apoptosis. Their activity is influenced by tumor molecular background, can synergize with conventional hormone therapies, and is maintained in therapy-resistant settings, highlighting their translational potential for the treatment of hormone-driven and resistant cancers.

## 5. Conclusions

The findings of this study highlight the promising potential of the hybrid compounds SJ028, SJ064, and SJ078 as innovative anticancer agents, particularly against hormone-driven and therapy-resistant tumors. The rational molecular hybridization approach enabled the generation of molecules with significant cytotoxic effects, capable of inducing apoptosis, promoting cell cycle arrest, causing DNA damage, and modulating key components of the PI3K/AKT signaling pathway. These compounds maintained their efficacy in three-dimensional spheroid models and in cell lines with acquired resistance to hormone therapies while also demonstrating additive and synergistic effects when combined with clinically used hormone therapies, underscoring their potential translational relevance. Moreover, results from AKT silencing experiments and molecular docking analyses support the hypothesis that this kinase is a principal molecular target of the hybrids. Collectively, these data not only advance the understanding of the mechanisms underlying the action of these novel compounds but also provide a robust foundation for future preclinical studies, positioning SJ028, SJ064, and SJ078 as promising candidates for the development of more effective, resistance-resilient, and clinically combinable cancer therapies.

## Figures and Tables

**Figure 1 pharmaceutics-17-01470-f001:**
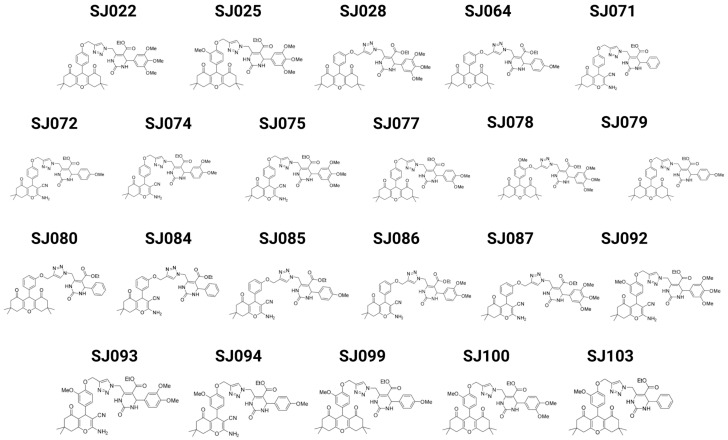
Chemical structures of the 22 xanthene- and pyran-based hybrid compounds evaluated in this study. These molecules were designed and synthesized as potential anticancer agents and were investigated for their antitumor activity throughout the work.

**Figure 2 pharmaceutics-17-01470-f002:**
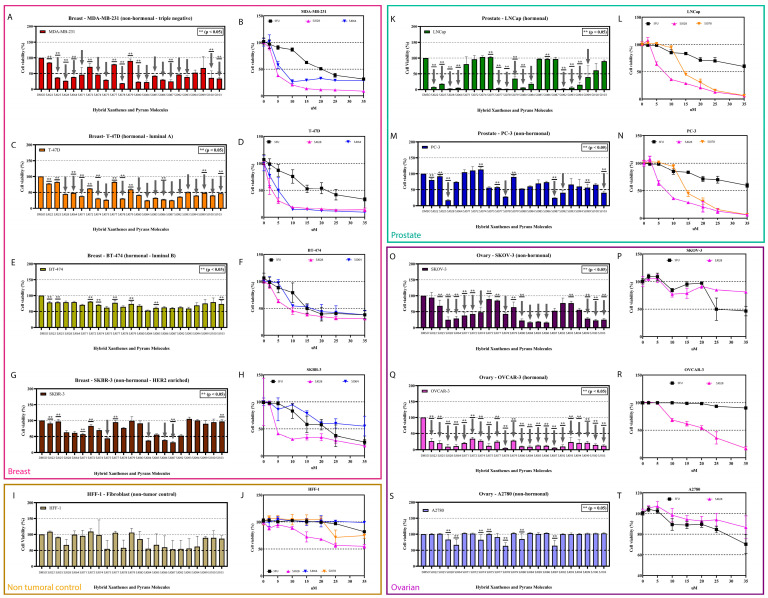
Cell viability of tumor and non-tumor cell lines after one- and minimum five-dose screenings of hybrid molecules. One-dose (10 μM, 72 h) and five-dose (0–35 μM) screenings of hybrid molecules in breast (pink: T-47D, MDA-MB-231, BT-474, SKBR-3), prostate (green: LNCaP, PC-3), and ovarian (purple: OVCAR-3, SKOV-3, A2780) cancer cell lines, with fibroblasts (yellow: HFF-1) as controls. Bars show relative viability. Panels (**A**–**T**) correspond to one and five-dose layouts as indicated in the figure. ** indicates statistically significant differences (*p* < 0.05) compared to the DMSO control. Gray arrows highlight compounds that reduced cell viability by more than 50% at a single dose of 10 µM.

**Figure 3 pharmaceutics-17-01470-f003:**
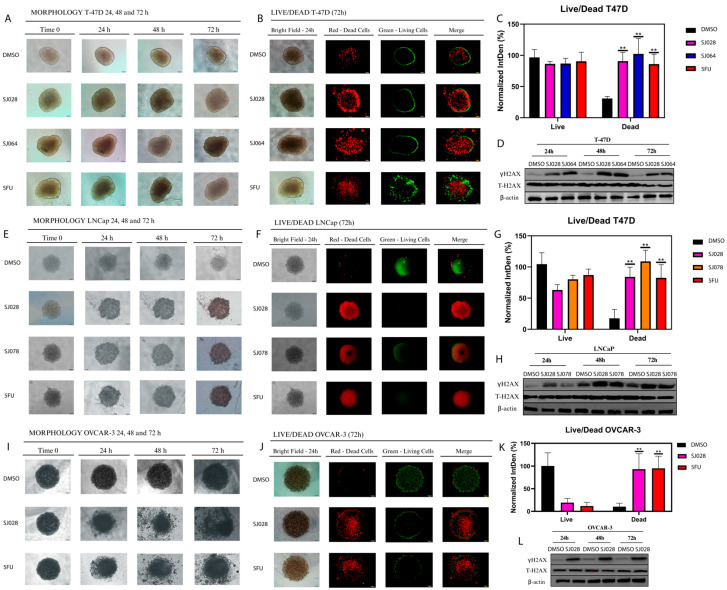
Morphological and molecular changes caused by hybrid molecules in 3D tumor models. Hybrid molecules induce morphological changes, increased cell death, and DNA damage in T-47D, LNCaP, and OVCAR-3 spheroids. T-47D was treated with SJ028 and SJ064, LNCaP with SJ028 and SJ078, and OVCAR-3 with SJ028. Micrograph of spheroid morphology captured at 10× magnification using bright field microscopy (**A**,**E**,**I**), live (green) and dead (red) fluorescence (**B**,**F**,**J**), quantitative viability analysis (**C**,**G**,**K**), and γH2AX phosphorylated (**D**,**H**,**L**) after 72 h of treatment. ** indicates statistically significant differences (*p* < 0.05) compared to the DMSO control.

**Figure 4 pharmaceutics-17-01470-f004:**
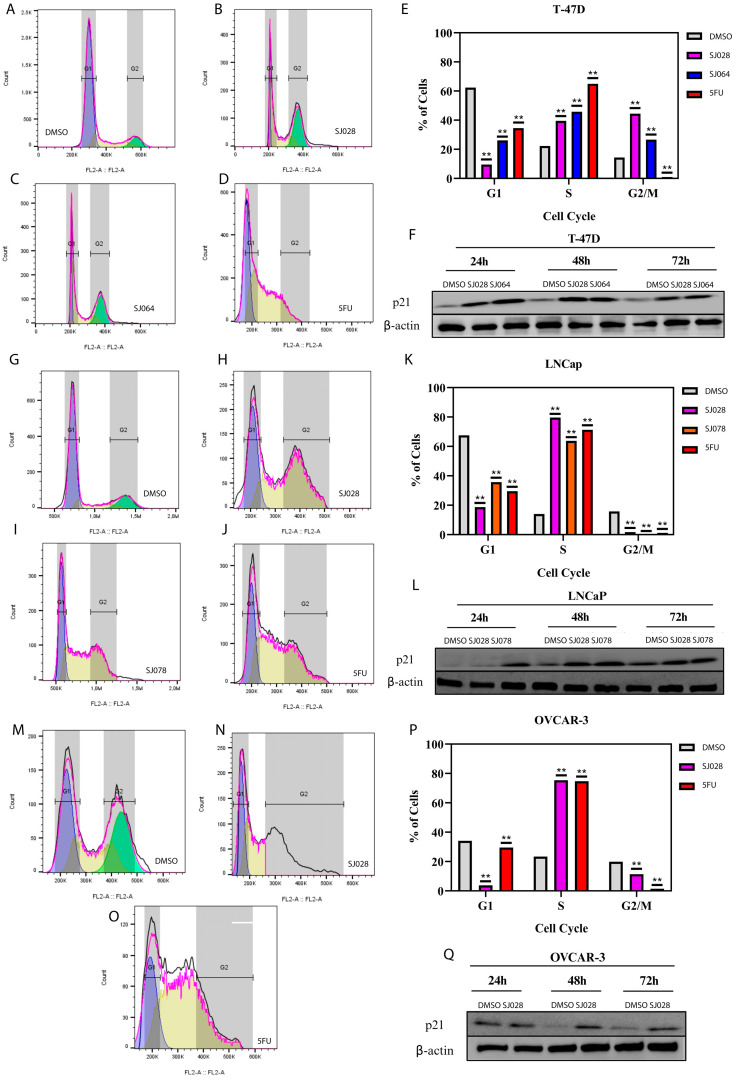
Analysis of cell cycle phases and inhibitors in 3D spheroid tumor models. The area under the curve is divided into phases: the blue peak represents cells in the G1 phase, the central yellow area represents the S phase (synthesis), and the green peak represents cells in the G2/M phase. The corresponding phases are demarcated by the shaded gray areas on the graph. (**A**–**D**) Cell cycle population of T-47D cells after treatment with DMSO (control), SJ028, SJ064, or 5-FU; (**G**–**J**) LNCaP cells treated with DMSO, SJ028, SJ078, or 5-FU; (**M**–**O**) OVCAR-3 cells treated with DMSO, SJ028, or 5-FU; (**E**,**K**,**P**) represent the quantitative percentages of cells arrested in each phase of the cell cycle for T-47D, LNCaP, and OVCAR-3, respectively; (**F**,**L**,**Q**). Western Blots of p21 expression after 48 h of treatment. Beta-Actin was used as control. ** indicates statistically significant differences (*p* < 0.05) compared to the DMSO control.

**Figure 5 pharmaceutics-17-01470-f005:**
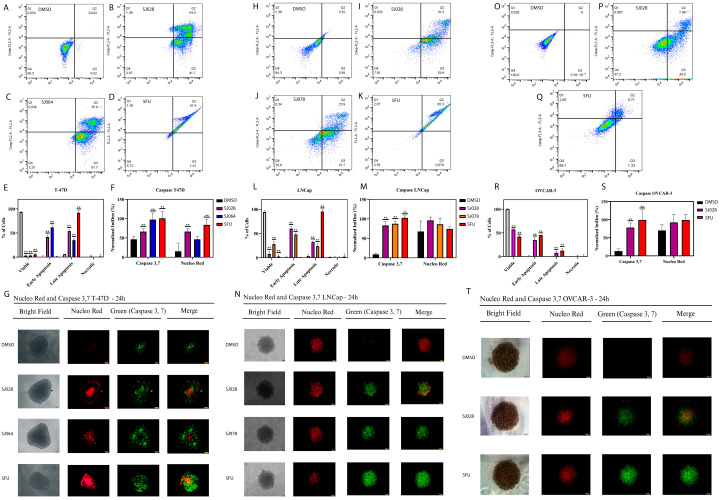
Analysis of cell death by cytometry and fluorescence in 3D spheroid tumor models. The color intensity corresponds to the relative density of cellular events in a region of the plot, ranging from blue (low density) to green (medium density), yellow (high density), and red (maximum density). (**A**–**D**) T-47D apoptosis measured as early, late, and necrotic populations following treatment with DMSO (grey), SJ028, SJ064, or 5-FU; (**H**–**K**) LNCaP apoptosis after DMSO (grey), SJ028, SJ078, or 5-FU; (**O**–**Q**) OVCAR-3 apoptosis after DMSO (grey), SJ028, or 5-FU; (**E**,**L**,**R**) quantitative analysis of apoptotic populations in T-47D, LNCaP, and OVCAR-3, respectively; (**G**,**N**,**T**) scale bar: 100 nm. Caspase 3/7 activation and NucRed staining; (**F**,**M**,**S**) Graphical quantification of caspase activity and NucRed intensity in the respective cell lines, highlighting significant induction of apoptosis and necrosis by the hybrid molecules compared to controls (*p* < 0.05), confirming cytotoxic efficacy in 2D and 3D models. ** indicates statistically significant differences (*p* < 0.05) compared to the DMSO control.

**Figure 6 pharmaceutics-17-01470-f006:**
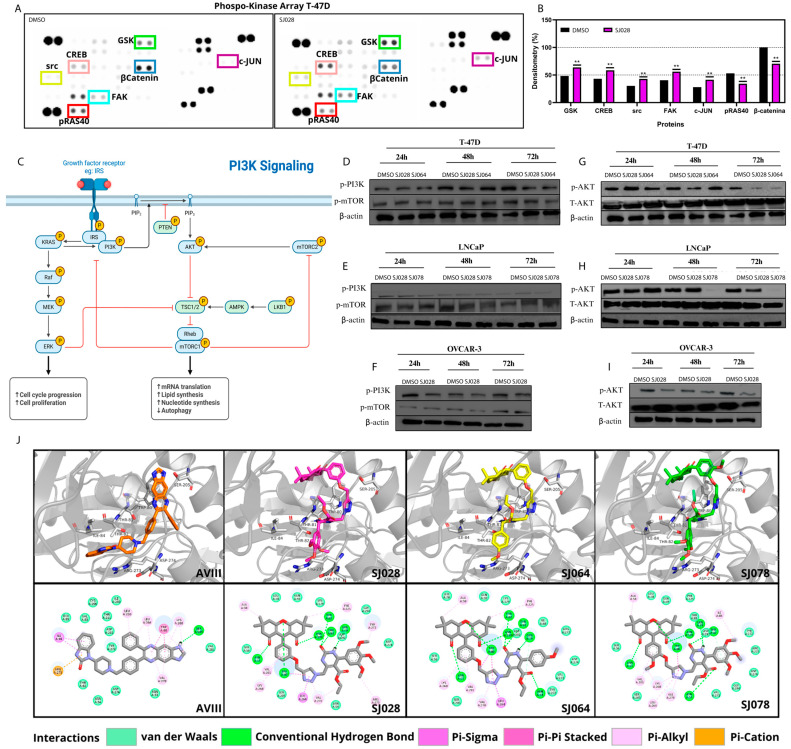
In vitro and in silico analysis of the molecular targets of xanthene-dihydropyrimidinone hybrid compounds. (**A**) Human phospho-kinase array of T-47D cells treated with SJ028 for 24 h or DMSO, showing global alterations in phosphorylation of multiple signaling proteins; (**B**) Densitometric analysis of altered dots (targets) after treatment with the hybrid molecule SJ028. (**C**) schematic illustration of PI3K/AKT/mTOR signaling pathway and its modulation by hybrid molecules; (**D**–**F**) Western Blots of phosphorylated PI3K, phosphorylated mTOR, and β-actin for T-47D, LNCaP, and OVCAR-3, respectively; (**G**–**I**) Western Blots of phospho- and total-AKT, and β-actin in T-47D, LNCaP, and OVCAR-3 treated with SJ028 or DMSO for 24 h. (**J**) Predicted binding modes and interaction profiles of inhibitors in the AKT1 allosteric site. The figure illustrates the three-dimensional (3D) predicted binding poses (top row) and corresponding two-dimensional (2D) ligand interaction diagrams (bottom row) for the reference inhibitor AVIII (orange sticks) and the novel compounds SJ028 (magenta sticks), SJ064 (yellow sticks), and SJ078 (green sticks). In the 3D views, AKT1 is depicted as a gray cartoon, with key interacting residues shown as white sticks. The 2D diagrams highlight the specific non-covalent interactions that stabilize each protein-ligand complex. ** indicates statistically significant differences (*p* < 0.05) compared to the DMSO control.

**Figure 7 pharmaceutics-17-01470-f007:**
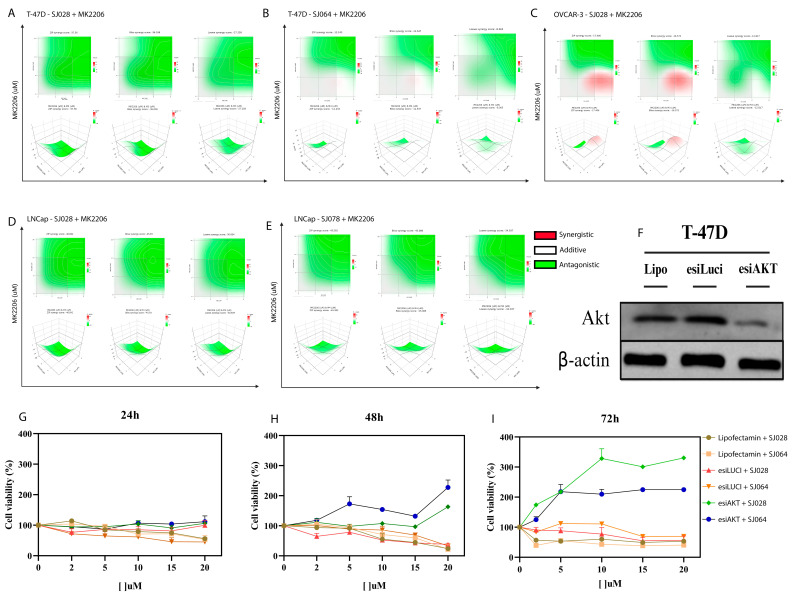
Pharmacological and genetic inhibition of AKT. (**A**–**E**) SynergyFinder analyses of hybrid molecules combined with the allosteric AKT inhibitor MK2206 in T-47D, LNCaP, and OVCAR-3. Green areas indicate antagonism, white areas additive effects, and red areas synergy across ZIP, Bliss, and Loewe models; (**F**) Western Blot confirming AKT silencing by esiRNA in T-47D cells; (**G**–**I**) Real-time cell viability measurements at 24, 48, and 72 h.

**Figure 8 pharmaceutics-17-01470-f008:**
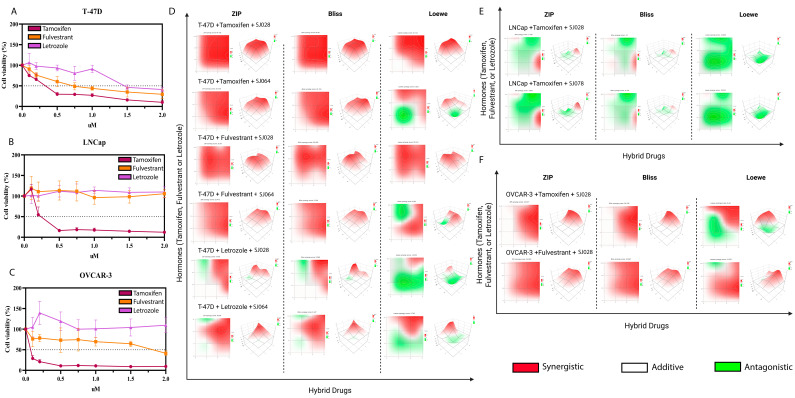
Hormone therapies synergize with hybrid molecules in hormone-driven cancer lines. (**A**–**C**) Dose–response viability curves of T-47D, LNCaP, and OVCAR-3 treated with tamoxifen, fulvestrant, or letrozole. (**D**–**F**) Synergy analyses between hybrid molecules and hormone therapies in T-47D, LNCaP, and OVCAR-3. Green areas indicate antagonism, white areas additive effects, and red areas synergy across ZIP, Bliss, and Loewe models.

**Figure 9 pharmaceutics-17-01470-f009:**
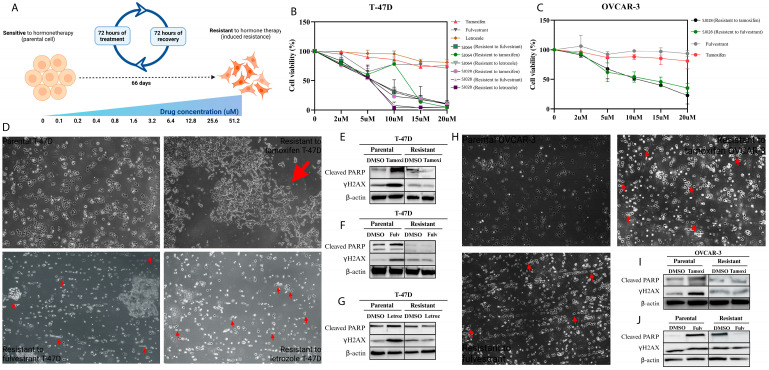
Establishment, characterization, and treatment of acquired antihormone resistance models. (**A**) Schematic overview of resistance induction protocols in T-47D and OVCAR-3 cells; (**B**,**C**) cell viability assays; (**D**,**H**) morphological characterization (scale bar: 100nm); (**E**–**G**,**I**,**J**) Western Blot analyses for molecular validation of hormone therapy resistance. The red arrow indicates the morphological changes visible in the resistant cell lines compared to the parental ones.

**Table 1 pharmaceutics-17-01470-t001:** IC_50_ values and selectivity indices (SI) of selected xanthene-dihydropyrimidinone hybrid compounds in tumor cell lines.

Hybrid Molecules				
Breast Cancer Cells	SJ028 (IC_50_ μM)	SJ064 (IC_50_ μM)	(SI) SJ028	(SI) SJ064
T-47D	3.44 ± 1.42	5.27 ± 1.08	>10.17	>6.64
BT-474	4.32 ± 1.05	5.62 ± 0.97	>8.10	>6.23
MDA-MB-231	3.94 ± 0.99	7.19 ± 0.88	>8.88	>4.87
SKBR-3	7.08 ± 0.87	36.11 ± 1.54	>4.94	<2
**Hybrid Molecules**				
**Prostate Cancer Cells**	**SJ028 (IC_50_ μM)**	**SJ078 (IC_50_ μM)**	**(SI) SJ028**	**(SI) SJ078**
PC-3	8.92 ± 1.36	15.56 ± 1.41	>3.92	>2.34
LNCaP	12.08 ± 1.27	17.38 ± 1.08	>2.90	<2
**Hybrid Molecules**				
**Ovarian Cancer Cells**	**SJ028 (IC_50_ μM)**		**(SI) SJ028**	
OVCAR-3	18.72 ± 0.95		2.19	
SKOV-3	>35		<2	

## Data Availability

The original contributions presented in this study are included in the article/Appendix A. Further inquiries can be directed to the corresponding authors.

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
