# Peer review of "Hybrid Dihydropyrimidinones Targeting AKT Signaling: Antitumor Activity in Hormone-Dependent 2D and 3D Cancer Models"

_pharmaceutics, 2025, doi:10.3390/pharmaceutics17111470_

Round 1

Reviewer 1 Report

Comments and Suggestions for Authors

Review pharmaceutics-3934774

The manuscript entitled „Hybrid Dihydropyrimidinones Targeting AKT Signaling: Antineoplastic activity in Hormone-Dependent 2D and 3D Cancer Models” submitted by Tejada A.H. et al. includes the results of multi-team research focused, at the beginning, on the selection of the most sensitive cancer cells among the panel of nine breast, prostate and ovarian cancer cell lines. Further, the screening for cytotoxicity of a series of 22 hybrid xanthene-dihydropyrimidinones was performed, and then, a whole range of assays analysing cytotoxic and proapoptotic activities of tested compounds with the use of breast (T-47D), prostate (LNCaP) and ovarian (OVCAR-3) cancer cell lines. The mechanism of the cytotoxicity of tested hybrid compounds was studied with selected 3 most active compounds, named SJ028, SJ064 and SJ078. The effect of these compounds on PI3K/AKT/mTOR pathway was estimated with Western blot and their affinity to  AKT binding site was studied by molecular docking. Moreover, hormone- nonresponsive cell lines were generated and it was concluded that the hybrids were also active in these cells. In responsive cell lines, the additive and synergistic effect with the corresponding antiestrogenic treatment was observed.

The manuscript (both the main text and the supplement material) is extensive and presents the results of many studies performed. However, in my opinion, some experiments are described too superficially or incorrectly, omitting essential details needed for final interpretation of results.

The main critisism

Abstract: The cancer cell lines were not selected as a result of in silico screening (line27). Ten cell lines (a panel of 9 cancer cell lines + 1 fibroblast cell line) were tested in vitro what should be included in the abstract. Why synergistic effect  is not mentioned in the abstract (Section 3.3. line 467)?

Introduction: Characteristic of biological activities of xanthene and pyrene derivatives could be included in the Introduction (positions 18-22 in References).

Materials and Methods. The synthesis of some compounds is not known. Reference cited (14. Santos, S.J., et al., Chromene-dihydropyrimidinone and xanthene-dihydropyrimidinone hybrids: design, synthesis, and antibacterial and antibiofilm activities) concerns only 11 xanthene hybrids. Next 11 hybrids which synthesis is described in this article [14], are chromene- dihydropyrimidinone, what is very confusing!

Line 141: …the most promising hybrid molecules…write please which of them.

Line 226: Tamoxifen, fulvestrant and letrozole are not hormones; they are hormone antagonists and as such are used in hormonal therapy.

Subsection 2.12. Acquired antihormonal resistance models. Line 237: Why hydroxytamoxifen were used instead of tamoxifen?

Figures: In general, authors should take care of better legibility of figures. Very small descriptions make difficult to see the results properly.

Figure 1. I understand the method five-dose screening, however it is misleading that there are more points on the graphs. Maybe it is better “minimum five-dose screening”?

Figure 5. This figure is overloaded. Wouldn’t it be better to split it into two? Because there is lack of showing the structures of selected hybrids SJ028, SJ064 and SJ078, it would be desirable to insert these structures to the figure presenting in silico analysis.

Table 6 (Suplementary)

IC₅₀ values for the hybrid drugs in parental cell lines should be included in this Table to make possible to compare the values without looking for relevant data in the main text. What was the concentration of tamoxifen, fulvestrant, or letrozole added to the respective cell cultures?

Results:

Line 326-328: Why the authors give two values (G1 population, S-phase fraction and G2/M population in %) comparing them with that one of DMSO? Moreover, in the Figure 3 the description what ** means is lacking. In general, statistical methods are not described in the manuscript. 

Figure 4: Results concerning caspase 3/7 are unreadable!

Line 385: Not only SJ028 but all tested compounds were docked to the AKT active site.

Line 409: The tested hybrids are not novel, they were (partly?) published earlier [14].

Line 411: The values given in the Table 5 (Supplementary) are different from these in the text. Moreover, -14,3 kcal/mol is lower than  -12.3 kcal/mol and it is not a control value.

Lines 426-431: This part of the text is unclear, and the Figure 6 G-I does not explain the conclusion formulated.

Discussion:

What the authors mean as “translational potential”?

Line 576: Please, show how it was proved that all AKT isoforms were knocked down.

References are not prepared according to MDPI recommendations. Position 14 is not completed. Biovia Discovery Studio also should be included in References.

Comments on the Quality of English Language

I am not a native speaker. Mostly, language is understandable. Unclear wording is indicated in the review.  

Author Response

Comment-1. Abstract: The cancer cell lines were not selected as a result of in silico screening (line27). Ten cell lines (a panel of 9 cancer cell lines + 1 fibroblast cell line) were tested in vitro what should be included in the abstract. Why synergistic effect  is not mentioned in the abstract (Section 3.3. line 467)?

Response 1 : We thank the reviewer for this thoughtful observation. Indeed, in the previous version of the manuscript, the in silico component of the study had not been explicitly described. We have now clarified this aspect in the revised version. The in silico analysis, focusing on drug-likeness and physicochemical properties according to Lipinski’s Rule of Five, was performed for the most active hybrid molecules, SJ028, SJ064, and SJ078, and the corresponding data have been added to the Supplementary Material. 

Furthermore, the Abstract was updated to specify that the compounds were tested in vitro in a panel of ten cell lines (four breast, three ovarian, two prostate cancer lines, and one non-tumorigenic fibroblast line), followed by the selection of T-47D, OVCAR-3, and LNCaP cells for detailed mechanistic analyses. The revised Abstract also now includes mention of the synergistic effects observed between the hybrid compounds and hormone therapies in breast and ovarian cancer models, as well as the antagonistic interactions with the AKT inhibitor MK2206 across all tumor subtypes. These modifications ensure that both the computational and experimental aspects of the study are accurately represented.

Comment-2. Characteristic of biological activities of xanthene and pyrene derivatives could be included in the Introduction (positions 18-22 in References).

Response 2: We thank the reviewer for this valuable suggestion. The Introduction section was revised to include a concise discussion of the biological activities of xanthene and pyran derivatives, as recommended. The new paragraph now emphasizes their cytotoxic, antiproliferative, and pro-apoptotic properties across breast, prostate, and ovarian cancer models, as well as their potential in overcoming resistance mechanisms through multitarget modulation. We added the following sentence: "The present study aimed to systematically evaluate a library of 22 xanthene–dihydropyrimidinone hybrid compounds for their antitumor potential, elucidate their putative cellular targets, and characterize their mechanisms of action in hormone-driven cancer models. The novelty of our work lies in testing 11 compounds that were previously evaluated for antimicrobial activity [14] for the first time in the context of cancer, alongside 11 entirely new hybrids never before assessed. This approach allows us to explore the antitumor potential of both previously known and novel hybrids, while investigating their mechanisms of action and possible simultaneous targeting of signaling and cell viability pathways."

Comment-3. The synthesis of some compounds is not known. Reference cited (14. Santos, S.J., et al., Chromene-dihydropyrimidinone and xanthene-dihydropyrimidinone hybrids: design, synthesis, and antibacterial and antibiofilm activities) concerns only 11 xanthene hybrids. Next 11 hybrids which synthesis is described in this article [14], are chromene- dihydropyrimidinone, what is very confusing!

Response 3: We thank the reviewer for pointing out this important issue and apologize for the lack of clarity regarding the synthesis and structures of the 22 hybrid compounds used in this study. To address this, we have added Figure 1 in the Materials and Methods section, which presents the chemical structures of all 22 molecules evaluated in the work.

Additionally, we have included further details on the synthesis of the compounds to clearly distinguish the xanthene–dihydropyrimidinone hybrids from the chromene–dihydropyrimidinone hybrids cited in reference [14]. These additions ensure that the manuscript now provides a complete and unambiguous description of the chemical structures and synthetic origins of all compounds studied. All those informations were added in  2.1. Hybrid Compounds subsection. 

Comment-4.Line 141: …the most promising hybrid molecules…write please which of them.

Response 4: We agree the reviewer and have added the names of all hybrid molecules used in westerns blotting experimentation. "After treatment of tumor cells with the most promising hybrid molecules (SJ028, SJ064, SJ078) of each type of tumor, protein extracts were prepared." at Line 237. 

Comment-5. Tamoxifen, fulvestrant and letrozole are not hormones; they are hormone antagonists and as such are used in hormonal therapy.

Response 5 : We thank the reviewer for this clarification and have corrected the text as suggested, at 375 Line. 

Comment-6. Acquired antihormonal resistance models. Line 237: Why hydroxytamoxifen were used instead of tamoxifen?

Response 6: We thank the reviewer for this question. Hydroxytamoxifen, the active metabolite of tamoxifen, was used in our experiments because it directly interacts with the estrogen receptor and provides a more consistent and reproducible effect in in vitro models, compared to the prodrug tamoxifen. This explanation has been added to the text at 386 Line. 

Comment-7. In general, authors should take care of better legibility of figures. Very small descriptions make difficult to see the results properly.

Response 7: We appreciate the reviewer's suggestion and would like to inform you that all figures were provided in a resolution higher than 1200 dpi in the manuscript submission system. Unfortunately, figures quality of word template is reduced at the submission manuscript. 

Comment-8. I understand the method five-dose screening, however it is misleading that there are more points on the graphs. Maybe it is better “minimum five-dose screening”?

Response 8: We thank the reviewer for this suggestion and have corrected the text in the Methods section as well as in Figure 1 to use “minimum five-dose screening” to clarify the methodology, specifically at Lines 398 and 477. 

Comment-9. Figure 5. This figure is overloaded. Wouldn’t it be better to split it into two? Because there is lack of showing the structures of selected hybrids SJ028, SJ064 and SJ078, it would be desirable to insert these structures to the figure presenting in silico analysis.

Response 9: We thank the reviewer for this valuable suggestion. To address the lack of structural representation without overloading Figure 5, we added Figure 1 in the Methods section, showing the chemical structures of all hybrid molecules used in the study. In addition, the in silico analysis has been included in the Supplementary Material for better organization and clarity.

Comment-10. Table 6 (Suplementary) IC₅₀ values for the hybrid drugs in parental cell lines should be included in this Table to make possible to compare the values without looking for relevant data in the main text. What was the concentration of tamoxifen, fulvestrant, or letrozole added to the respective cell cultures?

Response 10: We thank the reviewer for this valuable comment. The reviewer is correct, the concentrations of tamoxifen, fulvestrant, and letrozole used during the induction of resistance were not fully described in the original version. This information has now been included in the Materials and Methods section, specifying that the hormone therapies were added at the same concentrations used in the Cell Five-Dose Screening (0, 2, 5, 10, 15, 20, 25, and 35 μM) for IC₅₀ determination. Furthermore, the IC₅₀ values of the parental cell lines for the hybrid compounds have been incorporated into the Supplementary Table 8 to facilitate direct comparison with the resistant cells.

Comment-11. Line 326-328: Why the authors give two values (G1 population, S-phase fraction and G2/M population in %) comparing them with that one of DMSO? Moreover, in the Figure 3 the description what ** means is lacking. In general, statistical methods are not described in the manuscript.

Response 11: We thank the reviewer for this valuable observation. This experiment was performed in two independent biological replicates, which is why two values are shown in the text for the G1 population, S-phase fraction, and G2/M population when compared to DMSO.  All values in the manuscript text were corrected for their means. In addition, the statistical methodology has now been included in the Materials and Methods section, and the description of ** has been added to the figure legends.

Comment-12. Figure 4: Results concerning caspase 3/7 are unreadable!

Response 12: We thank the reviewer for this valuable comment. We have improved the quality of the Figure by increasing the resolution to 1200 dpi. 

Comment-13. Line 385: Not only SJ028 but all tested compounds were docked to the AKT active site.

Response 13: We thank the reviewer for this valuable observation. We have corrected the text to indicate that all tested compounds, not only SJ028, were docked to the AKT active site at Line 407. 

Comment-14. Line 409: The tested hybrids are not novel, they were (partly?) published earlier [14].

Response 14: We thank the reviewer for this comment. We have corrected the text to clarify that, although some of the tested hybrids were previously reported [14], the novelty of our work lies in demonstrating their antitumor activity. We also erased the word novel at line 759. 

Comment-15. Line 411: The values given in the Table 5 (Supplementary) are different from these in the text. Moreover, -14,3 kcal/mol is lower than  -12.3 kcal/mol and it is not a control value.

Response 15 We thank the reviewer for pointing this out. Indeed, the values in the text were incorrect. We have corrected them in the manuscript to match the correct values presented in Supplementary Table 7.

Comment-16. Lines 426-431: This part of the text is unclear, and the Figure 6 G-I does not explain the conclusion formulated.

Response 16.  We thank the reviewer for this comment. We have re-written this part of the text to make it clearer and better explain the results. We added the following sentence at LIne 775: "Importantly, esiRNA silencing of AKT isoforms abolished the cytotoxic effects of the hybrid compounds. Figure 7G–I shows that in AKT-silenced cells, SJ028 and SJ064 no longer reduced cell viability, whereas control cells (positive AKT expression) exhibited a strong viability reduction (~70%)."

Comment-17. What the authors mean as “translational potential”?

Response 17: We thank the reviewer for this comment. By “translational potential,” we refer to the ability of the hybrid compounds to demonstrate consistent cytotoxic activity across different preclinical models, including 2D cultures and 3D spheroids, as well as favorable selectivity indices (IC₅₀ < 35 µM for SJ028, SJ064, and SJ078) that are relevant for oncological drug development stages. These findings suggest that the compounds could potentially be further developed toward clinical applications.

Comment-18. Line 576: Please, show how it was proved that all AKT isoforms were knocked down.

Response 18: We thank the reviewer for this comment. The simultaneous knockdown of all AKT isoforms was achieved by transfecting cells with a pool of esiRNAs specific for AKT1, AKT2, and AKT3 (esiAKT1/2/3; Sigma-Aldrich). Knockdown efficiency was confirmed by Western blotting using a pan-AKT primary antibody (Akt (pan) (C67E7) Rabbit mAb #4691, Cell Signaling), which recognizes all three AKT isoforms. We have also added this information to the Materials and Methods section describing the esiRNA-mediated silencing.

Comment-20. References are not prepared according to MDPI recommendations. Position 14 is not completed. Biovia Discovery Studio also should be included in References.

Response 20: We thank the reviewer for this observation. We have corrected all references to comply with MDPI guidelines, completed reference 14, and added the citation for Biovia Discovery Studio in the reference list.

Reviewer 2 Report

Comments and Suggestions for Authors

Hormone dependent tumors are one of the important challenges of antitumor therapies. The excellent high-level research results described in the paper can significantly contribute to the development of this problem area. At first glance, my reviewer's opinion is that the work can be published without changes. However, despite my search for the structures of the significant hybrid molecules at the beginning of the thesis, I could not find them.
I think that the work is unacceptable without them. In particular, the structure-activity relationship would have been worth a few sentences.
If the authors replace this, the rest of the paper is fine. However, it cannot be accepted without it.

So: the structures should be placed at the beginning of the work, and some figures can be included in the Supplementary section.

Author Response

Comment-1. In particular, the structure-activity relationship would have been worth a few sentences.

Response 1 : We sincerely thank the reviewer for their positive assessment and valuable feedback that contributes to improving our manuscript. We fully agree that including the structural information of the hybrid molecules is essential for a better understanding of the study. Following this suggestion, we have added a paragraph at the beginning of the Hybrid Compounds section within the Methodology, where we now provide a detailed description of the molecular structures and discuss their antimicrobial activity. This addition also reinforces the structure–activity relationship and strengthens the scientific context of our findings.

Comment-2. The structures should be placed at the beginning of the work, and some figures can be included in the Supplementary section.

Response 2: We thank the reviewer for this helpful suggestion. To address it, we have added Figure 1 in the main text, which presents the chemical structures of the 22 hybrid molecules. In addition, we have included Table 1 in the Supplementary Material, listing all 22 compounds along with their corresponding molecular weights. These additions aim to improve the clarity and accessibility of the structural information presented in the manuscript.

Reviewer 3 Report

Comments and Suggestions for Authors

Referee report on the manuscript written by Amanda Helena Tejada et al. and entitled: “Hybrid Dihydropyrimidinones Targeting AKT Signaling: Anti-tumor Activity in Hormone-Dependent 2D and 3D Cancer Models”.

The manuscript presents biological activity results with combination of in silico study based on docking method. The manuscript is interesting, but before it will be ready for publication, the Reviewer would like the Authors to clarify some issues listed below.

The Authors wrote: “This study systematically screened the antineoplastic potential of 22 xanthene-dihy-dropyrimidinone hybrid compounds and investigated their putative cellular targets and mechanisms of action”, but they could write more about the main aims of the study. They should stronger underline the novelty of the manuscript. In addition, they could provide some examples of compounds exhibiting discuss in the manuscript biological activity.

In section 2 entitled Materials and Methods the Authors present the description of biological assays, docking procedure and the selected hybrid compounds. In the Reviewer’s opinion the Authors should change the division of the subparagraphs. They should start with the compounds presentation, later there should be the description of all biological procedures applied, the last part should contain the in silico study description.

The Authors refer to their previous work Ref. 14th in the current study, but the reference is not completed: “14. Santos, S.J., et al., Chromene-dihydropyrimidinone and xanthene-dihydropyrimidinone hybrids: design, synthesis, and antibacterial and antibiofilm activities” therefore it is impossible to check and compare the previous and the current study. This must be clarified and corrected.

The Reviewer would like to know how the Authors estimated their findings of the cell viability of tumor and non-tumor cell lines results (Figure 1) in accordance with other studies concerning different compounds or a reference structure. The question is valid for all biological assays. The Authors discuss the xanthene derivatives, but they do not provide an exact comparison to show that the proposed compounds are better. For instance they should prepare some tables and convince the reader that the reported compounds exhibit more promising biological activity. This will be valuable and the reader could see immediately the quantitative differences. Are the compound soluble only in DMSO ? The Authors should comment in the current study on the solubility of the proposed compounds as well as on Lipinski rule of 5. 

In the SI data the Authors wrote that the AutoDock software was used for the docking study. However, in the manuscript body there is written that they used AutoDock Vina v. 1.2.0. This must be clarified, because if they used two programs this must be clearly written, because the idea of the programs is the same, but technical details are different (binding affinity, scoring function). Why the Authors in the SI data provided binding affinity only for SJ028 hybrid molecule, but they omitted other molecules presented in the manuscript body (e.g. SJ064 and SJ078). It would be interesting to see the differences between them. Moreover, in line 411 they discuss binding affinities of other compounds in accordance with the control molecules. The data for the control molecule as well as other molecules discussed should be added to the manuscript body or to the SI. The Authors wrote: “The novel compounds SJ028, SJ064, and SJ078 exhibited strong predicted binding affinities of −12.3, −12.5, and −11.1 kcal/mol, respectively. Although slightly lower than the control, these values indicate a high binding potential for the AKT1 allosteric site (Supplementary Figure 1 and Supplementary Table 5)”, but as we can see in Figure 1 the redocing results for the control molecule are presented and in Table 5 the binding affinity only for SJ028.

In the References section, authors should pay attention to the doubled numbering of references. In addition, they should check carefully the Author Guidelines for the journal, because the References provided in the manuscript are almost prepared according to the journal requirements.

Author Response

Comment-1. They could write more about the main aims of the study. They should stronger underline the novelty of the manuscript. In addition, they could provide some examples of compounds exhibiting discuss in the manuscript biological activity.

Response 1: We thank the reviewer for this insightful comment. To address it, we have revised the end of the Introduction to better emphasize the main aims and the novelty of our work. The updated text now clearly highlights the innovative aspect of the hybrid design and strengthens the overall scientific context of the study. In addition, to address the reviewer’s suggestion regarding examples of compounds exhibiting biological activity, we have included new information in the Introduction, specifically at line 132 and 139-147. These modifications enhance the clarity, novelty, and scientific contribution of the Introduction section.

Comment-2. The Authors should change the division of the subparagraphs. They should start with the compounds presentation, later there should be the description of all biological procedures applied, the last part should contain the in silico study description. 

Response 2: We thank the reviewer for this helpful suggestion. Following their recommendation, we have reorganized the Materials and Methods section to present the information in the suggested order: first, the description of the synthesized hybrid compounds; next, the biological procedures applied; and finally, the in silico (molecular docking) studies. This new structure improves the logical flow and overall clarity of the new version of the manuscript.

Comment-3. The Authors refer to their previous work Ref. 14th in the current study, but the reference is not completed: “14. Santos, S.J., et al., Chromene-dihydropyrimidinone and xanthene-dihydropyrimidinone hybrids: design, synthesis, and antibacterial and antibiofilm activities” therefore it is impossible to check and compare the previous and the current study. This must be clarified and corrected.

Response 3: We thank the reviewer for noticing this issue. The reference has been corrected and now includes the complete citation details for that previous work. Additionally, we have inserted a new Figure 1 in the main text presenting the chemical structures of all the hybrid compounds used in this study, which facilitates comparison between the previous and current works.

Comment-4. The Reviewer would like to know how the Authors estimated their findings of the cell viability of tumor and non-tumor cell lines results (Figure 1) in accordance with other studies concerning different compounds or a reference structure. The question is valid for all biological assays. The Authors discuss the xanthene derivatives, but they do not provide an exact comparison to show that the proposed compounds are better. For instance they should prepare some tables and convince the reader that the reported compounds exhibit more promising biological activity. This will be valuable and the reader could see immediately the quantitative differences. Are the compound soluble only in DMSO ? The Authors should comment in the current study on the solubility of the proposed compounds as well as on Lipinski rule of 5.

Response 4: We thank the Reviewer for the valuable and constructive comments. In response, we have revised the manuscript and supplementary material to address all points raised as follows:

- Selection of the most promising compounds based on Selectivity Index (SI):

The values of the Selectivity Index (SI), presented in Supplementary Table 3, were used to identify the most promising hybrid molecules. Compounds that exhibited higher SI values relative to the other molecules tested in the study were considered more selective toward tumor cells versus non-tumor cells and therefore prioritized for further evaluation. This approach allows a direct visualization of relative potency and selectivity, supporting the rationale for compound selection throughout the study.

- Methodology (one- and five-dose screenings):

We clarify that the experimental procedures for the one- and five-dose screenings were adapted from the NCI-60 Screening Methodology of the National Cancer Institute, a widely recognized protocol for evaluating antiproliferative activity. These procedures have also been applied in peer-reviewed studies, such as Design, synthesis and evaluation of quinazoline-chalcone hybrids as inducers of cell-cycle arrest and apoptosis in breast cancer via DNA damage and CDK2/ATR inhibition, supporting the robustness and reproducibility of our assays. This framework ensures that our cell viability results are comparable with literature data.

STRINGHETTA, Giulia Rodrigues; MASS, Eduardo Bustos; GOMES, Izabela Natalia Faria; et al. Design, synthesis and evaluation of quinazoline-chalcone hybrids as inducers of cell-cycle arrest and apoptosis in breast cancer via DNA damage and CDK2/ATR inhibition. European Journal of Medicinal Chemistry Reports, v. 13, p. 100250, 2025. DOI: 10.1016/j.ejmcr.2025.100250.

- Lipinski’s rule of five:

The physicochemical properties of the hybrid compounds were analyzed according to Lipinski’s rule of five. The numerical values used for this analysis, including molecular weight (MW), number of hydrogen bond donors (HBD) and acceptors (HBA), predicted partition coefficient (Consensus Log P), and number of violations, are provided in Supplementary Table 2.

Both theses informations were used to classify the most promising hybrids molecules 

- Solubility and formulation:

All compounds were prepared as 10 mM stock solutions in DMSO and diluted in culture medium to achieve a final DMSO concentration below 1% (v/v), which did not affect cell viability. As expected for xanthene and dihydropyrimidinone hybrids, aqueous solubility is limited but sufficient for in vitro assays. The limited aqueous solubility of the xanthene-dihydropyrimidinone hybrid compounds is mainly due to their structural and physicochemical properties. These molecules contain large, rigid aromatic systems (xanthene and heterocyclic rings) and hydrophobic regions that reduce favorable interactions with water molecules. Although polar functional groups such as carbonyls and nitrogen atoms are present, their number and accessibility are insufficient to fully compensate for the extensive hydrophobic surface. Additionally, the relatively high molecular weight and lipophilicity (high Log P) further decrease water solubility, favoring solubility in organic solvents. Consequently, compounds were initially dissolved in DMSO to prepare stock solutions before dilution in culture medium for biological assays.

- Supplementary Material:

To provide further quantitative support and drug-likeness data, we have included two new tables in the Supplementary Material:

- Supplementary Table 2: Lipinski’s Rule of Five parameters for the molecules SJ028, SJ064, and SJ078.

- Supplementary Table 3: Selectivity indices (SI) of all compounds that passed the one-dose screening and were further evaluated in five-dose step. SI values across different cancer cell lines indicate the relative cytotoxicity toward tumor versus non-tumor cells, with higher SI reflecting greater selectivity for cancer cells. This table served as the basis for selecting the most promising hybrid molecules for further evaluation (SJ028, SJ064, SJ078).

These revisions strengthen the comparative and physicochemical context of our study, provide methodological transparency, and highlight the relevance of our findings in the broader literature.

Comment-5. In the SI data the Authors wrote that the AutoDock software was used for the docking study. However, in the manuscript body there is written that they used AutoDock Vina v. 1.2.0. This must be clarified, because if they used two programs this must be clearly written, because the idea of the programs is the same, but technical details are different (binding affinity, scoring function). Why the Authors in the SI data provided binding affinity only for SJ028 hybrid molecule, but they omitted other molecules presented in the manuscript body (e.g. SJ064 and SJ078). It would be interesting to see the differences between them. Moreover, in line 411 they discuss binding affinities of other compounds in accordance with the control molecules. The data for the control molecule as well as other molecules discussed should be added to the manuscript body or to the SI. The Authors wrote: “The novel compounds SJ028, SJ064, and SJ078 exhibited strong predicted binding affinities of −12.3, −12.5, and −11.1 kcal/mol, respectively. Although slightly lower than the control, these values indicate a high binding potential for the AKT1 allosteric site (Supplementary Figure 1 and Supplementary Table 5)”, but as we can see in Figure 1 the redocing results for the control molecule are presented and in Table 5 the binding affinity only for SJ028.

Response 5: We thank the reviewer for this valuable comment. We apologize for the inconsistency. All docking studies were performed using AutoDock Vina v. 1.2.0. The mention of AutoDock in the Supplementary Information was a typographical error, which has now been corrected.

In addition, we acknowledge the reviewer’s concern regarding the reported binding affinities. We have now incorporated all relevant compounds, including SJ028, SJ064, SJ078, and the control molecule, into a table of binding affinities in the Supplementary Information (Supplementary Table 7). The manuscript text has also been updated to explicitly report and discuss the binding affinity values for all compounds consistently. These revisions ensure that the comparisons among novel hybrids and control molecules are transparent and complete, and they allow the reader to fully appreciate the differences in predicted binding potential for the AKT1 allosteric site. We believe that these changes fully address the reviewer’s comment and improve the clarity and completeness of the docking results.

Comment-6. In the References section, authors should pay attention to the doubled numbering of references. In addition, they should check carefully the Author Guidelines for the journal, because the References provided in the manuscript are almost prepared according to the journal requirements.

Response 6: We thank the reviewer for this observation. We have carefully revised the References section to correct the doubled numbering and to ensure that all references now fully comply with the journal’s Author Guidelines. The updated reference list has been thoroughly checked for consistency in formatting, order, and completeness.

Round 2

Reviewer 1 Report

Comments and Suggestions for Authors

I thank the authors for taking most of my comments into account. A few of my reservations still remain:

Response 2:

In the new paragraph the aim of the study is correctly presented.

However, the inclusion of literature data concerning characteristic of biological activities of xanthene and pyrene derivatives could justify the selection of chemical structures that have been designed, synthesized and subsequently their biological activities were estimated.

Response 3:

The chemical structures of tested hybrid compounds are very complex and additionally differentiated by substituents. That is good that the structures are now presented in the Figure 1. However, would be desirable to clarify which part of molecules is a core of a hybrid. That is why I would suggest to include the general formula of synthesized hybrids.

Important: Chemical identification of novel xanthene-dihydropyrimidinone hybrids (1H i 13C NMR, mass spectra) should be included at least in the supplement.

Response 5: Because the term of hormonal therapy is used in many places in the manuscript, it is desirable to explain the role of tamoxifen, fulvestrant and letrozole in hormonal therapy in the place where they are first mentioned.

Other comments

Line 128: Caption to the Figure 1: biological agent I would change to anticancer agents. Biological agent may be misunderstood as natural compound.

Line 351: add selectivity towards cancer cells

Line 370: “All SI values for the molecules tested in the one dose screening…” How it was possible to estimate IC50 in the one dose screening?

The line numbering is different in the available version of the manuscript so I couldn’t refer to some of the answers.

Comments on the Quality of English Language

I am not a native speaker. Mostly, language is understandable. Unclear wording is indicated in the review.  

Author Response

Comment 1: In the new paragraph the aim of the study is correctly presented. However, the inclusion of literature data concerning characteristic of biological activities of xanthene and pyrene derivatives could justify the selection of chemical structures that have been designed, synthesized and subsequently their biological activities were estimated.

Response 1: We thank the reviewer for this valuable suggestion. We agree that including literature data on the biological activities of xanthene and pyrene derivatives helps to better justify the selection of the chemical structures. Accordingly, we have added relevant references in the Introduction (lines 103-113) to provide additional context.

Comment 2: The chemical structures of tested hybrid compounds are very complex and additionally differentiated by substituents. That is good that the structures are now presented in the Figure 1. However, would be desirable to clarify which part of molecules is a core of a hybrid. That is why I would suggest to include the general formula of synthesized hybrids.

Important: Chemical identification of novel xanthene-dihydropyrimidinone hybrids (1H i 13C NMR, mass spectra) should be included at least in the supplement.

Response 2: We sincerely thank the reviewer for this insightful and important comment. We fully agree that highlighting the core structure of the hybrids and providing detailed chemical identification is essential for clarity and reproducibility. Accordingly, we have included in the Supplementary Material the general synthetic schemes (Schemes 1–4) that illustrate the core structures of the hybrid compounds and indicate their hybridization points. Furthermore, the complete spectral characterization of all xanthene–dihydropyrimidinone hybrids, including 1H and 13C NMR as well as HRMS data, has been added to the Characterization of Hybrid Compounds section at the end of the Supplementary Material. A reference to this addition has also been inserted in the main text (lines 125–131).

Comment 3: Because the term of hormonal therapy is used in many places in the manuscript, it is desirable to explain the role of tamoxifen, fulvestrant and letrozole in hormonal therapy in the place where they are first mentioned.

Response 3: We thank the reviewer for this valuable suggestion. We agree that it is important to clarify the role of the hormonal therapies used. Therefore, we have added the following paragraph from lines 278 to 285, at the first point where these drugs are mentioned:

“Tamoxifen acts as a selective estrogen receptor modulator (SERM), blocking estrogen binding and inhibiting ER-mediated transcriptional activity. Fulvestrant is a selective estrogen receptor degrader (SERD) that promotes receptor degradation and complete suppression of estrogen signaling. In contrast, letrozole is a nonsteroidal aromatase inhibitor that reduces estrogen synthesis by inhibiting the aromatase enzyme. Due to their distinct mechanisms of action, these drugs are widely used in hormone-dependent cancers and may also modulate estrogen signaling in ovarian and prostate models expressing ER [2].”

Comment 4: Line 128: Caption to the Figure 1: biological agent I would change to anticancer agents. Biological agent may be misunderstood as natural compound.

Response 4: We appreciate the reviewer’s suggestion. We have revised the caption in line 128, replacing “biological agent” with “anticancer agent”, as we agree that this term better reflects the intended meaning and avoids potential misunderstanding. Thank you for the valuable comment.

Comment 5: Line 351: add selectivity towards cancer cells

Response 5: We thank the reviewer for the suggestion. We have modified the paragraph in line 351 to emphasize the selectivity of the compounds towards cancer cells. The revised sentence now reads:

“[…] whereas none of the compounds reduced HFF-1 viability by more than 50%, indicating favorable initial selectivity towards cancer cells (Figure 2 I–J).”

We agree that this wording improves the clarity of the statement.

Comment 6: Line 370: “All SI values for the molecules tested in the one dose screening…” How it was possible to estimate IC50 in the one dose screening?

Response 6: We thank the reviewer for pointing out this mistake. We have corrected the sentence in line 370, replacing “one-dose screening” with “minimum five-dose screening”, as this was indeed an error in the previous version. The correct term should be five dose screening.

Reviewer 2 Report

Comments and Suggestions for Authors

The MS is acceptable in present form

Author Response

Comment 1: The MS is acceptable in present form
Response 1: We thank the reviewer for their positive evaluation and for considering our manuscript acceptable in its current form.